The effect of patellar taping combined with isometric strength training on pain, muscle strength, and functional performance in patients with patellofemoral pain syndrome: a randomized comparative study

Hasan Shahnaz sh.ahmad@mu.edu.sa
Department of Physical Therapy and Health Rehabilitation, College of Applied Medical Sciences, Majmaah University , Al Majmaah , Saudi Arabia
Mohr Maurice
Electronic publication date: 2025 May 12
Publication date: 2025
Volume: 13
Electronic Location ID: e19381
Received 2024 Oct 3; Accepted 2025 Apr 7
Copyright: ©2025 Hasan
Copyright year: 2025
Copyright holder: Hasan
License: This is an open access article distributed under the terms of the Creative Commons Attribution License, which permits unrestricted use, distribution, reproduction and adaptation in any medium and for any purpose provided that it is properly attributed. For attribution, the original author(s), title, publication source (PeerJ) and either DOI or URL of the article must be cited.
License URL: https://creativecommons.org/licenses/by/4.0/

Keywords: Patellofemoral pain syndrome, Knee pain, Strength, Maximum voluntary isometric contraction (MVIC), Pain, Numeric pain rating scale, Anterior knee pain scale (AKPS)

Funding: Deanship for Postgraduate Studies and Scientific Research R-2025-1709 This research was funded by the Deanship for Postgraduate Studies and Scientific Research through project number R-2025-1709 at Majmaah University, Majmaah, Saudi Arabia. The funders had no role in study design, data collection and analysis, decision to publish, or preparation of the manuscript.

==============================
Background

Patellar taping and quadriceps strengthening exercises are commonly used in physiotherapy to manage patellofemoral pain syndrome (PFPS). However, previous research has reported inconsistent findings regarding quadriceps strength gains at specific knee angles during strength training in individuals with PFPS.

Objectives

This study investigated the efficacy of patellar taping and quadriceps isometric strength training (quadriceps-IST) at 60° knee flexion on quadriceps strength, pain, and functional performance in female patients with PFPS.

Methods

A two-arm, parallel-group, randomized comparative design was employed. Sixty adult females with PFPS were randomly assigned to either the experimental group (n = 30), which received patellar taping combined with quadriceps strength training at 60°, or the control group (n = 30), which received placebo taping with the same training. Both interventions lasted six weeks. Pain intensity, quadriceps muscle strength, and functional performance were assessed using the numeric pain rating scale (NPRS), maximum voluntary isometric contraction (MVIC) at 60° knee flexion, single leg triple hop (SLTH) test, and anterior knee pain scale (AKPS). For within and between groups comparison, a Wilcoxon signed-rank test and a Mann–Whitney U tests was employed, with confidence interval (α) set at 95%.

Results

Within-group analysis showed significant improvements in NPRS and MVIC at 60° knee flexion, SLTH, and AKPS scores post-intervention (p < 0.05). Between-group comparisons revealed that the experimental group had significantly greater improvements in all outcomes at six weeks post-intervention. Additionally, the pre-to-post changes (i.e., mean difference scores) were larger in the experimental group compared to the control group, which confirmed the superiority of the experimental group over the control group.

Conclusions

The study demonstrates that patellar taping combined with quadriceps -IST at a 60° knee flexion improves pain, muscle strength, and functional performance compared to placebo taping combined with quadriceps-IST. These findings suggest that incorporating this combined approach may enhance rehabilitation outcomes for patients with PFPS, providing a valuable addition to clinical practice.

Trial registration

This study was registered prospectively in the ClinicalTrials.gov PRS under a trial identifier NCT05168332 and last updated date 15/03/2024.

Introduction

Patellofemoral pain syndrome (PFPS) is a broad term that covers various conditions affecting the patellofemoral joint (PFJ), including extensor mechanism dysfunction, patellar subluxation, and runner’s knee (Herrington, 2004). PFPS is usually characterized by physical and biomechanical changes within the joint, resulting in anterior knee pain during activities like running, squatting, or climbing stairs (Hasan et al., 2022). These changes may involve altered patellar tracking, muscle imbalances, and soft tissue restrictions, such as weakness in the quadriceps or tightness in the hamstring muscles, all contributing to joint dysfunction and pain (Alonazi et al., 2021; Kang, Hee & Ro, 2023). This condition tends to be more intense, persistent, and linked to early degenerative alterations in young people suffering from anterior knee discomfort (Crossley, 2014; Conchie et al., 2016a; Conchie et al., 2016b).

Patellofemoral pain syndrome is the primary contributor to knee pain among young adults and affects generally active individuals and athletes (McConnell, 1986). Worldwide, PFPS affects approximately 22.7% of the general population, with a possible increase during adolescence, impacting an estimated 20–30% of young individuals (Borschneck et al., 2021; Smith Benjamin et al., 2019). In such cases, some young patients diagnosed with patellofemoral pain syndrome (PFPS) may face a higher risk of anterior cruciate ligament (ACL) injury (Myer Gregory et al., 2015). Around 30.3% of individuals are reported to experience PFPS in Saudi Arabia (Aldharman et al., 2022).

PFPS has been linked to various factors, such as modifiable risk elements, including vastus medialis obliquus (VMO) weakness, patellar misalignment, and hypermobility (Nunes Gabriela et al., 2013; Dutton Rebecca, Khadavi Maryam & Michael, 2016; Kunene Sabelo, Serela & Taukobong, 2018; Songur et al., 2024). Among these, quadriceps muscle weakness is a critical factor, strongly correlating with the onset and recurrence of PFPS (Verschueren et al., 2020; Halabchi, Roya & Tahereh, 2013; Osorio Juan et al., 2013). Imbalances within the quadriceps muscles, especially between the vastus medialis oblique (VMO) and the vastus lateralis (VL) (Thongduang et al., 2022), can exacerbate patellar tilt (Lee & Hyun, 2013). Studies highlight the importance of strong VMO muscles in maintaining patellar stability and preventing PFPS. Delayed or insufficient activation of the VMO leads to patellar maltracking, impairing knee joint function (Wyndow et al., 2016).

The iPFRN released a consensus document in 2018 on PFPS management, highlighting a hip- and knee-centered exercise program as the most evidence-based approach for effectiveness (Collins Natalie et al., 2018).

The McConnell Taping technique was initially developed to correct abnormal patellar alignment by centralizing the patella and enhancing quadriceps strength, mainly targeting the engagement and strengthening of the VMO (McConnell, 1996; Earl Jennifer & Hoch Anne, 2011). It also provides short-term pain relief during movement and influences patellofemoral joint mechanics (Coqueiro et al., 2005; Shahnaz et al., 2012; Alba-Martín et al., 2015)

Previous studies have reported using different therapeutic exercises, quadriceps muscular strength training, patellar taping and other strengthening protocol at various knee flexion angles to improve quadriceps strength while managing the outcomes of PFPS (Steinkamp Lee et al., 1993). Most of them used a single intervention approach, but a few studies utilized a combined approach to manage similar conditions (Suter et al., 1998; Chan et al., 2001). One study utilized EMG Biofeedback supplementation training for patellar taping to improve quadriceps strength and patellar alignment among adult male athletes only (Paul & Balakrishnan, 2015; Honarpishe, Hossein & Gholamreza, 2015; Hasan et al., 2022). Another study used patellar taping as an adjunct to the EMG biofeedback strength training at various angles (300, 600, and 900) among young adult male athletes only (Hasan, 2023). To our knowledge, none of the studies reported utilizing a combined intervention approach, including patellar taping and IST to improve quadriceps muscle strength and patellar alignment while managing PFPS among young adult females.

Therefore, to fill this gap, this study aimed to determine the effect of patellar taping combined with IST at 60-degree knee flexion on pain, quadriceps muscle strength, and functional performance in young adult females with PFPS. This study hypothesed that at 60-degree knee flexion, the patellar tapping combined with isometric strength training will more beneficial than placebo tapping and isometric strength training on pain, quadriceps muscle strength, and functional performance in young adult females with PFPS. The study findings will provide a basis for developing an effective method for increasing quadriceps muscle strength and correcting patellar alignments as a treatment for PFPS.

Materials and methods

Study design

A two-arm parallel group randomized comparative study design was employed to evaluate a 6-week intervention. The participants were allocated equally (1:1) to both groups (n = 30/group).

Study settings

The participants with PFPS were first diagnosed by a consultant orthopaedic surgeon and referred for physiotherapy interventions to the orthopaedic rehabilitation lab, Majmaah University, Al Majmaah, Riyadh, Saudi Arabia. Second, a team of three skilled and experienced female physiotherapists, specializing in musculoskeletal disorders, McConnell taping techniques, and experienced over 18 years, screened the adult females with PFPS. Participants were enrolled in this study over a nine-month period, from December 30, 2022, to September 25, 2023, through the physical therapy clinic at Majmaah University and public social media announcements.

Study procedure

The participants were screened based on the study’s inclusion and exclusion criteria and recruited for this study. An online random number generator (RNG) (https://www.socscistatistics.com/utilities/random/default.aspx) was used to participants’ random allocation to the groups to minimizing selection bias and balancing potential confounding variables across groups, strengthen the study’s reliability and internal validity. Participants were assigned to one of the groups based on the generated random numbers (1–60). The sixty generated random numbers (such as 24, 1, 7, 33, 48, 11, 30, 3, 9, 55... and so on) were grouped into four segments, each consisting of 15 random numbers. The first, second, third, and fourth segments of random numbers (1–15 & 31–45 and 16–30 & 46–60 random numbers) were alternately allocated to the participants of groups 1 and 2. The participant’s study inclusion serial number was matched to the same number available in the generated random number segment that further belongs to either of the intervention groups. Two assistant physiotherapist who were kept blind to group allocation documented the study’s outcome measures. Evaluations were performed both pre- and post-intervention, with all female participants in each group successfully completing the trial session. The experimental group (Group 1) received patellar taping combined with Isometric quadriceps exercise at 600 knee flexion; the control group (Group 2) received Placebo patellar taping combined with Isometric quadriceps exercise at 600 knee flexion. All the participants were advised to avoid rigorous activities, using pain medication, and following concurrent treatments to strengthen the study’s internal validity. The study’s outcome measures included quadriceps muscle strength assessed at a 60-degree knee flexion, pain intensity, and functional performance. A CONSORT (2010) flow diagram (Fig. 1) depicts the study procedures, including enrollment, randomization, allocation, follow up, and analysis.

Figure 1 A CONSORT (2010) flow diagram depicts the study procedures.

Study participants

The study evaluated 150 adult females experiencing knee pain through phone consultations. The study included 60 adult active females with PFPS. Out of 60 participants, four from experimental Group 1 and 3 from the control Group 2 did not complete the study for various reasons, totalling seven who dropped out. They left the study before reaching to the initial two weeks of the study. Participants were included if they had experienced knee pain for at least eight weeks, worsened by certain activities, displayed a positive J sign, had a more symptomatic and misaligned knee in cases of bilateral involvement, and showed radiographic evidence of patellar malalignment. The exclusion criteria encompassed a history of knee fractures, patellar dislocation, knee deformities, flexion contractures, ligament or meniscal injuries, recent NSAID use, intra-articular injections, or knee osteoarthritis.

Outcomes (dependent variables)

Quadriceps muscle strength (Primary)

Quadriceps muscle strength was assessed using an ISOMOVE dynamometer positioned at 60 degrees of knee flexion. Data collection was performed using version 0.0.1 of the ISOMOVE system software (ISO-MANSW-IT Tecnobody). It is a valid assessment tool for measuring isometric peak torque (Mahapure Swapnil et al., 2021). Before testing, participants underwent a comprehensive training session to ensure they were fully equipped to use the equipment. Baseline measurements of muscle strength (Nm) were recorded pre-intervention prior to the start of treatment and again at six week’s post-treatment, providing a valid and reliable data set.

Participants were securely positioned using safety belts across the chest, thighs, and hips, with the shin pad placed 5.1 cm (2 inches) above the medial malleolus. Testing was conducted on the more symptomatic leg at a 60-degree knee flexion angle. Participants were instructed to keep their arms crossed over their chests and were verbally encouraged to apply maximum effort during 5-second contractions. Each test included three consecutive trials with a 2-minute rest between trials. The average of these trials was calculated and used for statistical analysis.

Pain intensity (Secondary)

The Numeric Pain Rating Scale (NPRS) is a well-established, reliable, and valid instrument for assessing pain intensity (Hasan, 2013; Alghadir et al., 2018). Participants rated their pain on a scale from 0 to 10, where 0 indicated no pain and 10 represented the highest pain level.

Knee function (Secondary)

Knee function was evaluated using the validated Kujala Anterior Knee Pain Scale (Kujala et al., 1993; Hamdan et al., 2019). The scale comprises 13 questions aimed at evaluating various aspects of PFPS, including activities such as walking, squatting, stair climbing, jumping, and running, as well as the presence of a limp, need for support, pain levels, and issues with abnormal or painful kneecap movement. The scores can range from 0 to 100, where the higher score represents superior knee functional capacity. Before administration to patients with PFPS, the Kujala questionnaires were translated into Arabic and reviewed by native speakers of Arabic who had a medical background and were knowledgeable about the original document.

Single-leg triple hop test (Secondary)

SLTH encompasses the landing and propulsion phases. It is an effective screening tool commonly used in clinical practice to recognize individuals at risk of knee injury and assess progress in patients with PFPS and ACL reconstruction (Dos Reis et al., 2015). The test is also a comprehensive tool for evaluating lower extremity muscle strength. It measures physical performance requiring significant muscular activity, making it an asset in sports medicine and physical therapy.

In this study, participants’ performance was assessed using the SLTH test. Participants stood on their symptomatic limb with their toes aligned with the starting line and performed three consecutive hops on the same limb. The distance from the starting line to the point where the back of their heel landed was recorded (Fig. 2). Each participant completed three trials with a three-minute rest interval between trials. The maximum distance (cm) achieved across the three trials was recorded as the baseline measure.

Figure 2 SLTH test measurements.

Figure 3 Taping application method.

Interventions

Patellar taping

The taping techniques, meticulously described by McConnell, were skillfully employed by the physiotherapist to address specific patellar malalignments, including lateral and anteroposterior tilt and medial patellar glide. To alleviate pressure on the infrapatellar fat pad, tape was applied below the patella, as shown in Fig. 3. Initially, hypoallergenic underwrap tape was used to prevent skin irritation and tension. This was followed by rigid McConnell tape to medially reposition the patella by pulling the skin and patella into alignment (McConnell, 1996; Hasan et al., 2022). For the placebo taping, a nonrigid hypoallergenic tape (placebo tape) was applied vertically from the center of the patella with the knee in a flexed position. Quadriceps strength training was conducted at 60 degrees of knee flexion without rigid patellar taping. The tape was removed after the strength training and outcome measurement or earlier if the participant experienced itching, redness, or discomfort.

With patellar taping, each participant was carefully instructed to perform IST for quadriceps muscle at 60° knee flexion and for hip adductors to strengthen quadriceps muscles as described below.

These exercises were chosen based on their ability to recruit maximal quadriceps muscle activity. Since no single exercise could achieve maximal activation of the quadriceps muscles, this study employed a combination of exercises to enhance the likelihood of recruiting optimal quadriceps muscle activity (Rutherford, Hubley-Kozey Cheryl & Stanish William, 2011).

Participants were instructed to complete three sets of stipulated exercises five times a week for six weeks. The procedure was thoroughly explained, and they were encouraged to focus on their quadriceps muscle activity, holding each contraction for 5 s followed by a 10-second rest.

Quadriceps-IST at 600 knee flexion

It was carried out performing the patellar taping combined with maximum voluntary isometric contraction exercises at 600 knee flexion. Participants were guided to use the ISOMOVE system (ISO-MANSW-IT Tecnobody) to perform isometric quadriceps contraction exercises at a 60-degree knee flexion angle. The regimen, carried out five days a week for six weeks, consisted of three sets of two isometric contractions, each lasting 5 s, with a 30-second rest between sets.

Isometric hip adduction exercise

Participants were instructed to lie on their backs with a pillow placed between their knees and press it firmly to activate the muscles through isometric hip adduction exercises. They performed three sets of 10 repetitions per session per day, five days a week for six weeks.

Isometric hip abduction

Participants were instructed to lie supine with their knees flexed at 60 degrees, and the therapist placed their hand at the lateral aspect of the distal end of the femur to resist the movement while doing the isometric activation of the hip. Participants were asked to resist the movement for 10 s in three sets of 10 repetitions per session per day, five days a week for six weeks.

Quadriceps-IST at 600 knee flexion with placebo patellar taping

The control group participants completed the same set of exercises using placebo patellar taping. A nonrigid hypoallergenic placebo tape was applied vertically from the center of the patella with the knee flexed. They conducted quadriceps-IST at 600 knee flexion without rigid patellar taping. Participants were instructed to complete three sets of exercises five times a week for six weeks.

Ethical considerations

The study received approval from the Institutional Review Board of the College of Applied Medical Science at Majmaah University, Saudi Arabia, under ethical approval number MUREC-Nov.20/COM-2022/18-2, dated 2/11/2022. This study was registered prospectively in ClinicalTrials.gov PRS under trial identifier NCT05168332. Participants were informed of research risks and benefits and provided written informed consent following the Helsinki Declaration’s standards.

Sample size

The effective sample size for this study was calculated based on parameters from a previous study by Alonazi et al. (2021) using the G*Power software (version 3.1.9.4). That study examined quadriceps muscle strength in individuals with patellofemoral pain syndrome (PFPS) across two groups and at two time points. Using a reported effect size of Cohen’s d = 0.80 (95% CI) and a desired statistical power of 0.80, the required sample size was estimated to be 24 participants per group. To account for an anticipated 20% attrition rate, the total sample size was increased to 30 participants per group for the current study.

Table 1 Descriptive details of the demographic characteristics and baseline outcomes scores within each group (N = 53; 95% CI).

Variables	Mean ± Standard deviation	Kolmogorov–Smirnov (95% CI)	
	Group 1 (n = 26)	Group 2 (n = 27)	Statistics	df	p-value	
Age (Years)	25.96 ± 0.96	26.22 ± 0.89	0.213	53	0.001*	
Height (cm)	156.96 ± 1.46	157.22 ± 1.58	0.185	53	0.001*	
Weight (kg)	67.69 ± 3.11	67.37 ± 2.98	0.185	53	0.001*	
BMI (kg/m−2)	27.48 ± 1.30	27.26 ± 1.02	0.138	53	0.014*	
NRPS	6.98 ± 0.61	7.06 ± 0.73	0.198	53	0.001*	
STN	109.85 ± 6.25	108.33 ± 6.23	0.135	53	0.018*	
SLTH	245.08 ± 24.13	248.41 ± 18.89	0.080	53	0.200	
AKPS	42.27 ± 6.26	43.96 ± 7.01	0.148	53	0.013*	
Notes.

* Significance value, if p < 0.05.

cm Centimetre

kg Kilogram

NPRS Numeric pain rating scale

STN Strength

SLTH Single leg triple hop

AKPS Anterior knee pain scale

Statistical analysis

Data was analysed using IBM software for social statistics SPSS v.28 (IBM SPSS v.28, Armonk, NY, USA). A Kolmogorov–Smirnov test of normality was employed for abnormal data distribution. Non-parametric tests, including a Wilcoxon signed rank test and Mann–Whitney U test, were performed to observe the efficacies of stipulated interventions on the scores of the outcome measures within and between the groups, respectively. Additionally, changes in mean scores from pre- to post-assessment (i.e., mean difference scores) for all outcomes in each group were analyzed to determined one group’s superiority over another. A confidence interval was set at 95% for all the statistical analyses.

Results

A total of 53 out of 60 female participants completed the trial. The demographic and clinical data are summarized in Table 1. The Kolmogorov–Smirnov test of normality revealed that the data for the baseline demographic characteristics and outcomes measures were non-homogeneously distributed in this study (Table 1). Therefore, non-parametric tests, including a Wilcoxon signed rank test and Mann–Whitney U test, were performed to observe the efficacies of stipulated interventions on the scores of the outcome measures within and between the groups, respectively. The mean scores and standard deviations for the demographic characteristics and baseline scores are presented in Table 1.

Table 2 Within-group comparison of the post-intervention mean scores of the outcomes with the baseline scores, using a Wilcoxon signed rank test (N = 53; 95% CI, 2-tailed).

Variables	Group 1 (N = 26)	Wilcoxon signed ranks test (95% CI)	
Timeline	N	Mean rank	Sum of ranks	Z-Statistics	p-value	
NPRS (Pr vs. Po)	26	13.50	351.00	−4.482	0.001*	
STN (Pr vs. Po)	26	13.50	351.00	−4.484	0.001*	
SLTH (Pr vs. Po)	26	13.50	351.00	−4.458	0.001*	
AKPS (Pr vs. Po)	26	13.50	351.00	−4.465	0.001*	
Group 2 (N = 27)	
NPRS (Pr vs. Po)		14.00	378.00	−4.561	0.001*	
STN (Pr vs. Po)		13.50	351.00	−4.463	0.001*	
SLTH (Pr vs. Po)		14.00	378.00	−4.543	0.001*	
AKPS (Pr vs. Po)		14.00	378.00	−4.545	0.001*	
Notes.

* Significance value, if p < 0.05.

NPRS Numeric pain rating scale

STN Strength

SLTH Single leg triple hop

AKPS Anterior knee pain scale

Within-group comparison of outcomes scores

Wilcoxon signed rank test for within-group comparison revealed a significant improvement (p <  − 05) for all the outcomes scores of NPRS, STN, SLTH, and AKPS within each group when post-intervention scores were compared with baseline scores (Table 2 & Figs. 4–7.

Between-group comparison of outcomes scores

For between-group comparison, the Mann–Whitney U test revealed a significant difference between the groups for all the outcomes scores of NPRS, STN, SLTH, and AKPS compared at a six weeks’ post-intervention (Table 3).

Additionally, the pre-to-post changes (i.e., mean difference scores) were larger in the experimental group compared to the control group, which confirmed the superiority of the experimental group over the control group in reducing pain intensity and improving muscle strength (STN), and functional performance (SLTH & AKPS) (Table 4 and Fig. 8).

Figure 4 Comparison of NPRS mean score (pre vs. post) among groups.

Figure 5 Comparison of STN mean score (pre vs. post) among groups.

Discussion

This study evaluated the impact of patellar taping combined with quadriceps-IST at a 60-degree knee flexion on pain intensity, quadriceps muscle strength, and functional performance in young adult female patients with PFPS. Participants in the experimental group underwent patellar taping combined with quadriceps IST at the specified angle, while the control group received placebo patellar taping alongside the same intervention.

Figure 6 Comparison of SLTH mean scores (pre vs. post) among groups).

Figure 7 Comparison of AKPS mean score (pre vs. post) among groups.

The study findings demonstrated overall improvements in the outcomes of pain, quadriceps muscle strength, and functional performance in both groups; however, the experimental group revealed more benefits over the control group in improving the same at six weeks of post-intervention.

Notably, all participants from an experimental group were eventually pain-free and fully functional. In contrast, participants from the control groups experienced residual pain and weakness in the quadriceps muscles. The results indicate the therapeutic efficacy of patellar taping combined with quadriceps-IST at a 60° knee angle, aligning with previous studies (Bockrath et al., 1993; Gilleard, McConnell & David, 1998). For instance, recent literature by Alonazi et al. (2021) supports this study finding by highlighting similar improvement using patellar taping, with electromyographic-biofeedback (EMG-BF) guided quadriceps-IST in young adult male athletes with PFPS over four weeks. Additionally, Alonazi et al. (2021) indicated that the quadriceps-IST at a 60° knee flexion achieved outperforming muscle strength than those training at 30 or 90 degrees. These findings suggest that the 60° flexion angle offers biomechanical advantages for quadriceps activation and strength development.

Previous studies indicated the proprioception and muscle fiber length-tension relationships as an underlying mechanism of benefits of patellar taping and isometric exercise (Aminaka & Gribble, 2008; Kochar Shantanu, Tejashree & Shrikant, 2024). Patellar taping may enhance proprioception by activating cutaneous mechanoreceptors, improving afferent input to the central nervous system, and reducing pain perception (Aminaka & Gribble, 2008). Concurrently, isometric exercises at specific knee angles may optimize muscle fiber length-tension relationships, enhancing strength and function (Kochar Shantanu, Tejashree & Shrikant, 2024).

Some researchers have suggested that properly applied patellar taping should result in at least a 50% reduction in pain during a step test (Gilleard, McConnell & David, 1998; McConnell, 1986) In a previous study, active patellar taping led to an average pain reduction of 47% during the step test, compared to only 10% with placebo taping. In the control group, repeating the step test without taping resulted in a mere 2% decrease in pain. These findings align closely with the pain reductions reported by other studies (Bockrath et al., 1993; Gilleard, McConnell & David, 1998; Wilson, Carter & Thomas, 2003).

In numerous studies, patellar taping has been widely investigated as a therapeutic intervention for PFPS, and demonstrated that it is effective in reducing the pain and functional improvement (Petersen et al., 2016; Barton et al., 2014; Sisk & Fredericson, 2020). The McConnell taping technique, which uses rigid tape to address lateral patellar glide, tilt, and rotation, is particularly effective in reducing pain during functional tasks like step-downs (Cowan, Bennell & Hodges, 2002). Additionally, individuals often use non-specific taping techniques, such as patellar glide and medial glide taping (Lack et al., 2018), which are intended to improve vastus muscle activation and coordination (Lee et al., 2012). These effects may be mediated through increased proprioceptive input and neural inhibition, collectively known as the nociceptive effect (Willy et al., 2019). While taping is effective for short-term management, it does not independently improve long-term muscle function. The Academy of Orthopaedic Physical Therapy has recently emphasized the need for a clinical practice guideline for managing PFPS (Willy et al., 2019). They recommend incorporating patellar taping into an exercise therapy regimen to provide immediate pain relief and enhance short-term outcomes (up to 4 weeks). However, they note that taping techniques lose effectiveness over time or with more intensive physical therapy interventions and do not support its use for improving muscle function (Willy et al., 2019). While patellar taping offers short-term benefits for individuals with PFPS, further research is required to better understand its long-term effects.

Table 3 Between-group comparison of pre-and-post-intervention mean outcomes scores, using a Mann–Whitney U test (N = 53; 95% CI).

Variables	Groups	N	Mean rank	Sum of ranks	Mann–Whitney U	Z- Statistic	Asympt. Sig. (2-tailed)	
NPRS-Pr	Group 1	26	25.56	664.50	313.500	−0.692	0.489	
Group 2	27	25.39	766.50	
NPRS-Po	Group 1	26	13.50	351.00	0.001	−6.283	0.001*	
Group 2	27	40.00	1,080.00	
STN-Pr	Group 1	26	29.44	765.50	287.500	−1.133	0.257	
Group 2	27	24.65	665.50	
STN-Po	Group 1	26	40.25	1,046.50	6.500	−6.135	0.001*	
Group 2	27	14.24	384.50	
SLTH- Pr	Group 1	26	26.23	682.00	331.000	−0.356	0.722	
Group 2	27	27.74	749.00	
SLTH- Po	Group 1	26	40.25	1,046.50	6.500	−6.133	0.001*	
Group 2	27	14.24	384.50	
AKPS-Pr	Group 1	26	25.46	662.00	311.000	−0.718	0.473	
Group 2	27	28.48	769.00	
AKPS-Po	Group 1	26	39.25	1,020.50	32.500	−5.712	0.001*	
Group 2	27	15.20	410.50	
Notes.

* Significance value, if p < 0.05.

Pr Pre intervention

Po Post intervention

NPRS Numeric pain rating scale

STN Strength

SLTH Single leg triple hop

AKPS Anterior knee pain scale

Table 4 Comparison of mean differences scores for the outcome variables (pre-& post measurements) within each group (N = 53).

Group 1 (n = 26); Mean ± SD	
Variables	Post intervention	Baseline	MD	SEM	
NPRS	1.38 ± 0.65	6.98 ± 0.61	5.60	0.173	
STN	154.38 ± 9.10	109.85 ± 6.25	44.53	1.874	
SLTH	356.88 ± 36.42	245.08 ± 24.13	111.80	8.500	
AKPS	80.38 ± 4.78	42.27 ± 6.26	38.11	1.037	
Group 2 (n = 27); Mean ± SD	
NPRS	4.26 ± 0.81	7.06 ± 0.73	2.80	0.869	
STN	122.37 ± 9.51	108.33 ± 6.23	14.04	8.877	
SLTH	268.19 ± 14.33	248.41 ± 18.89	19.78	14.991	
AKPS	69.52 ± 5.12	43.96 ± 7.01	25.56	6.333	
Notes.

MD Mean differences

SEM Standard Error of Mean

NPRS Numeric Pain Rating Scale

STN Strength

SLTH Single Leg Triple Hop

AKPS Anterior knee Pain Scale

Figure 8 Comparison of pre-to-post changes (mean differences score) for all outcome variables in each group.

The specific choice of the 60° knee flexion angle in this study is supported by biomechanical evidence. A study found that individuals who trained their quadriceps at a 60-degree knee flexion angle developed significantly greater quadriceps strength compared to those training at 30 or 90 degrees of knee flexion. Due to their distinct anatomical characteristics, the three superficial quadriceps muscle segments, including the vastus medialis obliquus (VMO), generate varying levels of muscle torque depending on the knee angle. Moreover, this causes knee angle changes to affecting muscle fibre excursion length (Pincivero et al., 2004). In another study by Shenoy, Mishra & Sandhu (2011) reported that the maximal quadriceps torque achieved at a 60° knee flexion angle.

In contrast, a study found that individuals who trained their quadriceps at a 60-degree knee flexion angle developed significantly greater quadriceps strength compared to those training at 30 or 90 degrees of knee flexion. Due to their distinct anatomical characteristics, the three superficial quadriceps muscle segments, including the vastus medialis obliquus (VMO), generate varying levels of muscle torque depending on the knee angle (Paul & Balakrishnan, 2015). These discrepancies may stem from participant characteristics or study methodology variations. Future research should address these inconsistencies to refine clinical recommendations.

Study limitations

This study has some limitations. First, it was limited to young adult female patients only with PFPS. Therefore, the findings cannot be generalized to male patients or older adults. Second, a follow-up was needed to assess the lasting effects of the intervention on pain intensity, muscle strength changes and functional performance. Further research is needed to investigate the long-term impact of corrective taping combined with isometric strength training at 60-degree knee flexion in individuals with PFPS. Additionally, studies should explore the relationship between pain, strength, and function in patients with PFPS to enhance our understanding of these interactions. We recommend comparing PFPS between activities involving an open-kinetic chain versus closed-kinetic chain exercises at different knee flexion angles. The long-term efficacy of combining patellar taping with isometric quadriceps training for PFPS rehabilitation. The long-term efficacy of combining patellar taping with isometric quadriceps training for PFPS rehabilitation.

Conclusions

The study demonstrated that patellar taping combined with quadriceps-IST at a 60° knee flexion angle significantly enhances pain relief, muscle strength, and functional performance compared to placebo taping combined with the same intervention. These findings highlight a promising approach for clinicians, practitioners, and physiotherapists, advocating for the integration of this technique into rehabilitation protocols to improve outcomes for patients with patellofemoral pain syndrome.

Supplemental Information

Supplemental Information 1 Raw Dataset

The data analysis output for the homogeneity tests of normal distribution and non-parametric analysis for comparing outcomes scores between and within groups.

Supplemental Information 2 CONSORT 2010 Checklist

Supplemental Information 3 Trial Protocol

The author is grateful to Mr. Shahid Hasan, Amir Iqbal, Raad I. Abdullah, and faculty members of the rehabilitation centre for their support and assistance in this research, including recruitment, group allocation, data analysis and interpretation.

Additional Information and Declarations

Competing Interests

Author Contributions

Human Ethics

Data Availability

Clinical Trial Registration

The authors declare there are no competing interests.

Shahnaz Hasan conceived and designed the experiments, performed the experiments, analyzed the data, prepared figures and/or tables, authored or reviewed drafts of the article, and approved the final draft.

The following information was supplied relating to ethical approvals (i.e., approving body and any reference numbers):

The Chair of Majmaah University for Research Ethics Committee, Saudi Arabia, approved this study under ethical approval number “MUREC-Nov.20/COM-2O22/18-2” dated 20/11/2022.

The following information was supplied regarding data availability:

The raw data is available in the Supplementary File.

The following information was supplied regarding Clinical Trial registration:

NCT05168332.

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
