# Peer review of "The effect of patellar taping combined with isometric strength training on pain, muscle strength, and functional performance in patients with patellofemoral pain syndrome: a randomized comparative study"

_PeerJ, doi:10.7717/peerj.19381_

## Round 0.1 · original submission · Major Revisions

· Academic Editor

Major Revisions

While I see the effort behind this study, there are important issues, which at the current stage do not allow publication of this article.

The reviewers have outline some important issues within your original submission, most notably:

1) The cited literature is not up-to-date. There is a 2018 consensus statement on current recommendations to treat PFP: Collins, N. J., (2018). British Journal of Sports Medicine, 52(18), 1170–1178. Please update the entire manuscript and refer to recent literature on this topic. There is a challenge by reviewer 3, that the investigations is missing novelty. While PeerJ does not judge submissions based on novelty, the rationale of your study must be described within the framework of current literature.

2) Strength testing: More information on the testing device is needed to judge reliability and validity. The device cannot be found online by searching for the information provided in the text. Please rely on published reliability and validity estimates if available.

Please also address the following issues:
1) What is the unit of measurement of your strength test? This is unclear and I have concerns about a 50% increase in quadriceps strength (!!) in a 6-week training period. E.g. group 1 increases from a value of 109 to 154 from baseline to post-intervention. Comparable studies have shown strength gains of ~ 10% (e.g. https://pmc.ncbi.nlm.nih.gov/articles/PMC5095944/). How is this possible? Connected to this: there is inconsistency in the text about the training frequency (5x in Line 253 vs. 3x in Line 262). Please clarify.

2) It is interesting that no drop-outs were reported. This seems unlikely for a group of sixty participants. Did everyone complete the study? Furthermore, the adherence to training is unknown. Therefore, the validity of the study is impossible to judge at this point. Critical questions remain: How "blinded" was the placebo group really? Most people may guess that a non-rigid tape will not have an effect. Did the placebo group actually achieve the same training frequency compared to the experimental group? How can the assistant physiotherapists who recorded the data be blinded to group allocation (Line 160)? I am sure, the different tapes will be obvious to them.

3) Statistical analysis: In randomized controlled studies, it is standard of practice to either use analysis of co-variance (ANCOVA) with the baseline value as covariate or analysis of variance (ANOVA) (see e.g. here https://www.sciencedirect.com/science/article/pii/S0895435606000813?via%3Dihub). Please explain why you chose to only analyze within-group changes and between-group differences at the post-measurement.

4) Apart from the content: Such randomized-controlled studies require extensive effort in recruitment, data collection, and analysis. Would it be appropriate to acknowledge supporters in the Acknowledgements section or by inclusion of additional authors?

·

Basic reporting

no comments

Experimental design

no comments

Validity of the findings

no comments

Additional comments

A graphical depiction of the trajectory of pain scores (NPRS) and functional measures (AKPS, SLTH) is missing. providing this for both groups across the study period would greatly enhance the reader's understanding of the intervention effects.
The validity and reliability of the ISOMOVE dynamometer for measuring quadriceps strength in PFPS patients require more thorough documentation.
The control of extraneous variables, such as participants' activity levels, pain medication use, and concurrent treatments, requires more attention to strengthen the study's internal validity.

Reviewer 2 ·

Basic reporting

The introduction section should start with the definition and scope of the problem given in line 60 and then the prevalence section should be given.

The sentence given in lines 69 and 72 may be connected to the previous paragraph.

In 2018, the iPFRN published a consensus document concerning PFP management that specified a hip- and knee-focused exercise regimen to be the approach with the highest level of evidence regarding its effectiveness (Collins NJ, Barton CJ, van Middelkoop M, et al. 2018 consensus statement on exercise therapy and physical interventions (orthoses, taping and manual therapy) to treat patellofemoral pain: recommendations from the 5th international patellofemoral pain research retreat, gold coast, Australia, 2017. Br J Sports Med. 2018; 52(18): 1170-1178. doi:10.1136/bjsports2018099397). Additionally, Songur et al. also reported the alignment changes in PFS and the effects of different banding on this alignment with MRI findings (Songur et al. The effects of different Taping methods on Patellofemoral alignment, pain and function in individuals with Patellofemoral pain: a randomized controlled trial. PM R 2023;16:474–84. doi:10.1002/pmrj.13067). Taking these into account, the introduction should be revised with more up-to-date sources.

“Performance” in line 122 and after should be written with lower case and the expression in parentheses should be deleted.

The phrase "patella taping" on line 119 and onwards should be written as "patellar taping".

Data regarding the findings in lines 176-178 should be given in the results section

Experimental design

After the hypothesis of the study, predictions about the possible outcomes of the study should be written and the purpose of the study should be given. In sentences written about the possible benefits of the study, it is not correct to start the sentences with "study’s findings" (lines 125-129).

References must be given for each method used in the "Materials and methods" section.

It should be noted that the Arabic version of the "Kujala Anterior Knee Pain Scale" was used, which is valid and reliable.

It should be explained why the hip adduction exercises in line 268 are being performed.

Although the demographic values and first measurement results of the participants in both groups given in Table 1 are very close to each other and the statistical comparison results are similar (p>0.05), it is not correct to state that the groups are not homogeneous according to only one normality test. For this reason, I recommend that other normality tests be performed and a decision be made and statistical analyses be performed according to the new results.

The sentence in lines 311-313 should be written as “Wilcoxon Signed Rank Test for within-group comparison revealed a significant improvement (p<-05) for all the outcomes scores within each group when post intervention scores were compared with baseline scores.”

Correct the table number in line 319.

All tables should be arranged in accordance with the current template of the journal.

There is no need for the word "however" in line 336.
The "groups" expression on line 338 should be used in singular.
There should be parentheses before the references on line 343.
More recent literature should be used in the discussion section. The findings of this current study should be interpreted. Too much general information content should not be included.
After the strengths of this study are given, limitations should be given.
Spelling errors (capitalization) and citation style in the references should be reviewed (e.g. line 492)

Validity of the findings

There is no need for the word "however" in line 336.

The "groups" expression on line 338 should be used in singular.

There should be parentheses before the references on line 343.

More recent literature should be used in the discussion section. The findings of this current study should be interpreted. Too much general information content should not be included.

After the strengths of this study are given, limitations should be given.
Spelling errors (capitalization) and citation style in the references should be reviewed (e.g. line 492)

Additional comments

Congratulations on your work. The last paragraph of the introduction should clearly state which gap in the literature this study fills, and in the discussion section, the findings from the study should be interpreted and their relationship with the hypothesis of the study should be explained.

·

Basic reporting

no comment

Experimental design

The Knowledge Gap which the author is trying to address is not a Gap at all. There are many quality existing studies including systematic reviews and consensus statements which have already addressed and proved these gaps.

Validity of the findings

The study is well-framed though it does'nt appears to be novel in present as already RCTs addressing these issues are conducted in past.

Additional comments

The study appears to be great work,however,there are some observations:
The PFPS and Chondromalcia are different entities ,Inclusion and exclusion criteria no where mentions about it.Clarity about it is required.

why only Isometric Hip Adduction Exercise:
were included when Hip Abduction Exercises were found to be superior in PFPS

---

## Round 0.2 · Major Revisions

· Academic Editor

Major Revisions

While some aspects have been clarified with this revision, critical points remain to be addressed by the authors to make this article acceptable for publication. Please see the editor comments (in green color) that have been added to your attached "response/rebuttal letter". Some are summarized below. Please provide a revised manuscript and point-by-point response of the editor and reviewer comments.

Most importantly: It is not acceptable to make up values for dropped-out participants for the post-measurement data set. These participants have to be removed and the whole manuscript must reflect, the actual sample size at the follow-up.

The added figures are of poor quality and contain errors (e.g. Figure 4). Please improve them according to the editor comment.

The statistical analysis needs revision. Individual pre-to-post changes in the outcome variables should be compared between the Groups using a t-Test or Mann-Whitney test depending on whether the pre-to-post changes are normally distributed. Further, the Cohen's d estimates in Table 4 are wrong and it is generally unclear how this analysis helps to answer the research question.

Reviewer 2 ·

Basic reporting

No comment.

Experimental design

No comment.

Validity of the findings

No comment.

Additional comments

1. The citation style of the following publication has been corrected as:
Songur Adil, Demirdel Ertuğrul, Kılıc Özlem, Akın Mustafa Emre, Alkan Afra, Akkaya Mustafa. 2024. The effects of different taping methods on patellofemoral alignment, pain and function in individuals with patellofemoral pain: A randomized controlled trial. PM & R 16(5): 474–484.

2. The citation style of the following publication should be corrected in accordance with the journal rules:
Hamdan, M., Haddad, B., Isleem, U., Hamad, A., Hussein, L., Shawareb, Y., Hadidi, F., Alryalat, S.A., Samarah, O., Khanfar, A. and Alzoubi, B. 2019. Validation of the Arabic version of the Kujala patellofemoral pain scoring system. Journal of Orthopaedic Science 24(2):290.293.

3. The spelling "tapping" in lines 110, 111 should be corrected to "taping".

·

Basic reporting

It is well framed presently & addressess the comments

Experimental design

After the revisions the desired aspects are clarified with relevance. However blinding can be a concern for true outcomes.

Validity of the findings

No comments

Additional comments

The queries appear to be relevantly addressed the current article can be published after minor revision.

---

## Round 0.3 · Minor Revisions

· Academic Editor

Minor Revisions

The manuscript has much improved through the revision of the figures, the consistent sample size after drop-outs, and the pre-to-post change analysis. A few smaller issues are to be resolved before publication:

1) Abstract: "The study demonstrates that patellar taping combined with quadriceps-IST at a 60° knee flexion improves pain, muscle strength, and functional performance compared to placebo taping" - ADD placebo taping combined with quadriceps-IST.

2) Line 125: I meant to only remove the word "reliable", because the cited study only assess validity.

3) Line 129: Why Nm-2?

4) Training frequency: I understood that the training goal was to achieve five sessions per week over six weeks. However, did you track the actual training adherence? How many sessions per week did each group achieve on average?

5) Sample size calculation (line 234): Which effect size of 0.08 do you refer here? Please state in the text.

6) Figures: Please replace Group 1 / Group 2 with their actual labels (experimental group, control group). Please indicate in the figure captions what the error bars represent (standard deviation, standard error, ..?).

---

## Round 0.4 · Minor Revisions

· Academic Editor

Minor Revisions

Thank you for clarifying some of the remaining points. What is still unclear is the sample size calculation. I do not understand your response that the stated effect size was based on the "previous/current study". Please explain in clear terms: 1) what type of effect size are you reporting here? (Cohen's f, eta-squared, Cohen's d?), 2) which study is this effect size based on? (you cite two studies in the text, Alonazi 2021 & Hasan 2022). Also please change "confidence interval 0.05" to "significance level 0.05". A confidence interval of 0.05 would correspond to a significance level of 0.95 and give you very different results.

---

## Round 0.5 · accepted · Accept

· Academic Editor

Accept

Congratulations on improving the manuscript to a level that allows publication in PeerJ.

During the proof-editing state: Please correct a mistake in the limitations section. There is a duplicate and the last sentences do not make sense by themselves. (Line 412-417: "We recommend comparing PFPS between activities involving an open-kinetic chain versus closed-kinetic chain exercises at different knee flexion angles. The long-term efficacy of combining patellar taping with isometric quadriceps training for PFPS rehabilitation. The long-term efficacy of combining patellar taping with isometric quadriceps training for PFPS rehabilitation.")